# Interferon-λ treatment accelerates SARS-CoV-2 clearance despite age-related delays in the induction of T cell immunity

Deanna M. Santer [1,5] ✉, Daniel Li [2,3,5], Yanal Ghosheh[3], Muhammad Atif Zahoor [3], Dhanvi Prajapati[1], Bettina E. Hansen[3], D. Lorne J. Tyrrell [4], Jordan J. Feld [2,3,5] & Adam J. Gehring [2,3,5] ✉

Interferons induced early after SARS-CoV-2 infection are crucial for shaping immunity and preventing severe COVID-19. We previously demonstrated that injection of pegylated interferon-lambda accelerated viral clearance in COVID-19 patients (NCT04354259). To determine if the viral decline is mediated by enhanced immunity, we assess in vivo responses to interferon-lambda by single cell RNA sequencing and measure SARS-CoV-2-specific T cell and antibody responses between placebo and interferon-lambda-treated patients. Here we show that interferon-lambda treatment induces interferon stimulated genes in peripheral immune cells expressing *IFNLR1*, including plasmacytoid dendritic cells and B cells. Interferon-lambda does not affect SARS-CoV-2-specific antibody levels or the magnitude of virus-specific T cells. However, we identify delayed T cell responses in older adults, suggesting that interferon-lambda can overcome delays in adaptive immunity to accelerate viral clearance in high-risk patients. Altogether, interferon-lambda offers an early COVID-19 treatment option for outpatients to boost innate antiviral defenses without dampening peripheral adaptive immunity.

SARS-CoV-2, the cause of COVID-19, has led to a pandemic that has resulted in >5.8 million deaths worldwide (https://coronavirus.jhu.edu/map.html). With the emergence of variants and the possibility of breakthrough infections, it is widely believed that SARS-CoV-2 will become an endemic virus[1]. Consequently, finding safe and effective treatments for COVID-19 remains a priority to prevent hospitalizations and deaths, and to expedite recovery in unvaccinated individuals or breakthrough infections.

Interferons (IFNs) are a crucial part of the innate antiviral immune response and drive the expression of a wide array of genes with antiviral and immunoregulatory properties, collectively known as interferon-stimulated genes (ISGs)[2]. Two families of IFNs contribute directly to the innate antiviral response at mucosal barriers in humans-

type I (eg. IFN-α, IFN-β) and type III (IFN-λs). The broad pleiotropic effects of ISGs can overcome antiviral resistance, making type I or III IFNs potential therapeutics for new and/or highly diverse viruses. Recent studies have found a link between severe COVID-19 and deficiencies in, or autoantibodies to, type I IFN, while stronger type I IFN responses have been associated with asymptomatic infection[3], highlighting the critical role of IFNs in disease evolution[4–6]. Like other viruses, SARS-CoV-2 encodes proteins to antagonize IFN responses[7–10], however supplementing the natural IFN response with IFN treatment has been found to be effective against the virus[7,10–13].

Type III IFNs act primarily at mucosal barriers through binding a unique heterodimeric receptor (IFN-λR1/IL-10RB) to promote innate antiviral immunity[14–17]. Restricted IFN-λ receptor distribution and a lack

[1]Department of Immunology, University of Manitoba, Winnipeg, MB, Canada. [2]Institute of Medical Science, University of Toronto, Toronto, ON, Canada. [3]Toronto Centre for Liver Disease, University Health Network, Toronto, ON, Canada. [4]Department of Medical Microbiology & Immunology, Li Ka Shing Institute of Virology, University of Alberta, Edmonton, AB, Canada. [5]These authors contributed equally: Deanna M. Santer, Daniel Li, Jordan J. Feld, Adam J. Gehring. ✉e-mail: Deanna.Santer@umanitoba.ca; Adam.Gehring@uhnresearch.ca

of IRF1 induction[18] result in influenza viral load decline without inflammatory side effects in mice treated with IFN-λ, whereas mice treated with Type I IFN show impaired survival[19]. However, the IFN-λ receptor is not solely restricted to epithelial barriers. We previously showed that functional type III IFN receptors are expressed on human immune cells, including B and T cell populations[20]. IFN-λ3 pre-treatment of human CD4 + T cells significantly inhibited human immunodeficiency virus-1 infection[20]. However, IFN-λ3 addition to peripheral blood mononuclear cells (PBMCs) also inhibited influenza vaccine-induced antibody production in vitro[21,22]. Recently, our group conducted a phase II placebo-controlled randomized trial of pegylated interferon-lambda (PEG-IFN-λ), a type III IFN, as therapy for mild-to-moderate COVID-19 in outpatients. PEG-IFN-λ treatment accelerated viral clearance compared to placebo without inflammatory side effects[11]. When controlling for baseline viral load, PEG-IFN-λ-treated patients were more likely than placebo patients to have undetectable viral load by day 7 (OR = 4.12, p = 0.029) and this effect was particularly pronounced in patients with a baseline viral load above $10^6$ copies/mL (OR = 6.25, p = 0.012). Whether the accelerated viral decline was related to direct antiviral properties of IFN-λ and/or enhancement of the SARS-CoV-2-specific immune response was not defined in the clinical study[11].

It is currently unknown how PEG-IFN-λ treatment affects virus-specific T and B cell responses in patients during an acute viral infection. Given the accelerated viral decline observed with therapy, we hypothesized that PEG-IFN-λ treatment induced a more robust SARS-CoV-2-specific specific T cell responses and dampened antibody production compared to placebo. We analyzed longitudinal T cell and antibody responses after therapy using single-cell RNA sequencing (scRNAseq), measurement of the SARS-CoV-2 specific antibody levels in plasma, and the magnitude and functionality of SARS-CoV-2-specific T cells. ScRNAseq confirmed in vivo responses to PEG-IFN-λ in specific peripheral immune cells, but treatment did not alter virus-specific adaptive immune responses. In fact, the antiviral effects of PEG-IFN-λ were observed despite a delayed T cell response in older patients at risk of more severe outcomes. Overall, PEG-IFN-λ treatment for COVID-19 is a promising early treatment that can accelerate viral clearance in patients with delayed T cell immunity.

## Results

### Specific peripheral blood immune cells are responsive to PEG-IFN-λ in vivo

Our prior in vitro experiments showed that subsets of peripheral blood immune cells express the IFN-λ receptor subunit IFN-λR1 and respond to IFN-λ exposure with up-regulation of ISGs[20]. To determine if a peripheral immune cell response to therapeutic administration of PEG-IFN-λ in vivo could be detected, we performed scRNAseq on 9 patients from the clinical study; 5 patients received PEG-IFN-λ compared to 4 placebo (control). ScRNAseq was performed to investigate expression of the IFN-λ receptor (IFNLR1/IL10RB) and to detect in vivo ISG responses in individual immune cell populations.

After filtering for high quality cells, we included 263,668 cells in our analysis; 146,408 cells from PEG-IFN-λ-treated and 117,260 from placebo patients. Clustering yielded 21 cell populations (Fig. 1A). Cell types were identified using canonical marker genes displayed in the dot plots and consistent with known cell types in peripheral blood (Fig. 1B). Expression of the heterodimeric IFN-λ receptor, IL10RB and IFNLR1, was visualized using feature plots. Expression of IFNLR1 was observed in specific immune populations, primarily B cells, plasmacytoid dendritic cells (pDCs) and granzyme B (GzmB)+ CD8 T cells (Fig. 1C). These populations were previously demonstrated to respond to IFN-λ in vitro[20,23]. IL10RB was ubiquitously expressed by immune cells (Fig. 1D).

We then determined which clusters expressed the highest level of each receptor component and what frequency of the cells had detectable receptor expression. pDCs expressed the highest level of

IFNLR1 (Fig. 1B, E) while monocytes, expressed the highest level of IL-10RB, but no IFNLR1, consistent with our previous work (Fig. 1B, F)[20]. To measure the response to PEG-IFN-λ, we developed a composite module score that factored in gene expression from 24 ISGs. (Supplementary Table 2). ISG module scores declined over time indicating that ISG expression was elevated at baseline in both PEG-IFN-λ and placebo patients as a result of the acute infection (Fig. 1G-J). However, upon PEG-IFN-λ treatment, pDCs maintained an elevated ISG response at D3 post treatment compared to D0 compared to placebo control patients (Fig. 1G). Monocytes, which do not express IFNLR1, served as an internal negative control for IFN-λ responsiveness and showed declining ISG expression in both IFN-λ and placebo patients (Fig. 1H). The frequency of IFNLR1 + cells in the other clusters was too low to observe a change in the ISG module score. Therefore, we enriched for IFNLR1 + cells from B cell clusters and measured the response to PEG-IFN-λ via the ISG module score, which demonstrated positive ISG responses at D3 for B cells in both clusters B cells 1 and Memory B cells when compared to D0 (Fig. 1I, J). Overall, these analyses demonstrated that IFNLR1 + immune cells in the peripheral blood responded, in vivo, to PEG-IFN-λ treatment in COVID-19 patients.

### Pegylated-IFN-λ treatment did not affect SARS-CoV-2-specific antibody levels compared to placebo

Antibody response to SARS-CoV-2 is a primary metric of protection following vaccination or prior exposure. To investigate the effect of PEG-IFN-λ on B cell responses, we quantified levels of total IgM, IgG and IgA in patient plasma (placebo; n = 11-12 and PEG-IFN-λ; n = 14-15 for each time point) at D0, D7, and D90 + . Total IgG levels were significantly higher at D0 and D7 compared to D90 + in both groups, indicating an increase in total IgG during early infection (Fig. 2A). We found there were no differences in total IgM, IgG, or IgA levels between placebo- and PEG-IFN-λ-treated patients at D0, D7, or D90 + post-enrollment (Fig. 2A). Additionally, total IgM decreased between D0 and D90 + in the PEG-IFN-λ patients, while both groups showed a decrease in total IgM between D7 and D90 + (Fig. 2A). For total IgA, there were significant differences in the placebo group, increasing between D0 and D7, and decreasing between D7 and D90 + (Fig. 2A).

To measure receptor binding domain (RBD)-specific IgG, IgM, and IgA antibody levels we utilized a spike RBD-specific ELISA protocol in patient plasma from each treatment group. Eight pre-pandemic plasma samples (from 2018-2019) were used as negative controls and displayed very little background (Fig. 2B, dotted lines). We observed a significant increase in RBD-specific antibodies in plasma from D0 to D7 for all subclasses. We also found no differences in RBD-specific IgG, IgM, or IgA levels at all three time points when comparing placebo- and PEG-IFN-λ-treated patients (Fig. 2B). At D90+, only RBD-specific IgG was still significantly elevated compared to D0 and D7, whereas both IgA and IgM antibody levels significantly decreased between D7 and D90+ (Fig. 2B). The decrease of RBD-specific IgA and IgM levels at D90+ was consistent between placebo and PEG-IFN-λ groups. RBD-specific IgG, IgA, and IgM levels correlated between patients at D7 (Supplementary Table 3). At D90+ when RBD-specific IgM and IgA antibodies were lower, there were no significant correlations between RBD-specific IgG, IgA, or IgM levels.

Overall, these results indicate that COVID-19 patients in both groups mounted RBD-specific antibodies above background and PEG-IFN-λ treatment did not inhibit or increase B cell antibody responses measured in plasma.

### Pegylated-IFN-λ treatment did not affect T cell responses compared to placebo

SARS-CoV-2-specific T cell responses towards the wild-type membrane (M), envelope (E), nucleocapsid (N), and spike (S) protein were measured in 38 clinical trial patients (placebo; n = 17 and PEG-IFN-λ; n = 21) at three time points (Table 1). We used an ex vivo three-colour

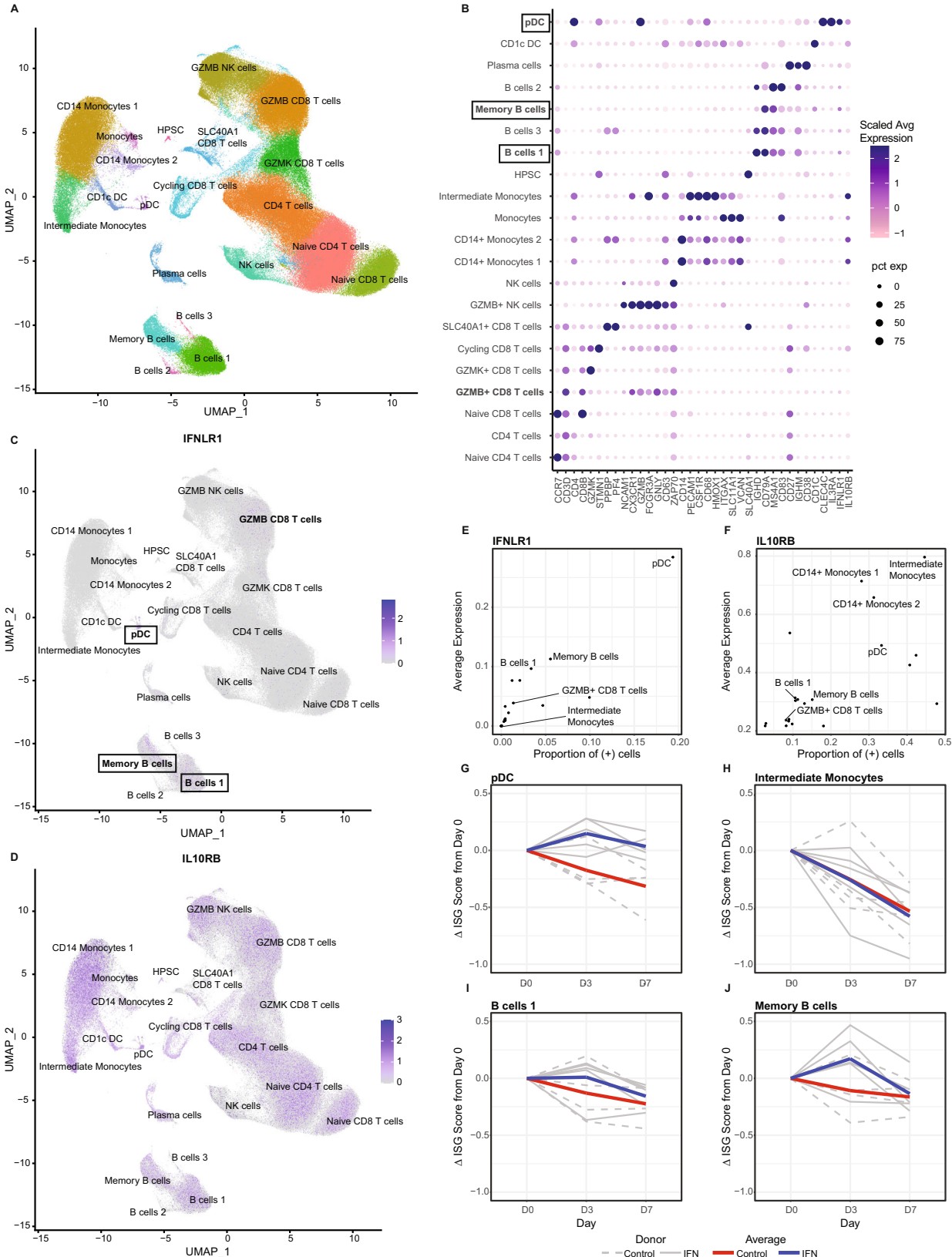

FluoroSpot assay detecting IFN-γ, IL-2, and granzyme B (GzmB) on patient PBMCs stimulated with SARS-CoV-2 peptides for 24 h. A response was considered positive when the average spot forming units (SFUs) of duplicate wells exceeded 2 times the individual's DMSO-stimulated negative control SFU count and greater than the mean negative SFU count from all patients. SFU counts were normalized by subtracting the background DMSO-stimulated SFU count of the individual patient time point.

More than 50% of patients showed positive T cell responses at D0, which was within 7 days of symptom onset and laboratory-confirmed SARS-CoV-2 infection (Supplementary Fig. 1). Robust IFN-γ and IL-2 responses were readily observed, whereas less than half of the patients

**Fig. 1 | Longitudinal scRNA-Seq analysis of cells from patients who were administered either PEG-IFN-λ or placebo. A** UMAP (Uniform Manifold Approximation and Projection) projection and clustering identified 21 clusters. UMAP_1 and UMAP_2 represent the first and second dimensions of the projected low dimensional graph, respectively. **B** Selected marker genes (*x*-axis) for each cluster (*y*-axis) to confirm their annotation. Color and size of the points reflect the level and the proportion of expression, respectively, for each gene in each cluster. *y*-axis labels reflect the annotation. Cell clusters with the highest IFNLR1 expression are highlighted by boxes. **C** *IFNLR1* and **D** *IL10RB* expression in clusters are displayed on feature plots. Proportion of cells which have non-zero (**E**) *IFNLR1* or (**F**) *IL10RB* expression in each cluster. Each point represents a cluster. *Y*-axis denotes the

average expression of the receptor in those cells in which it is expressed and *x*-axis denotes the proportion of these receptor-expressing cells in each cluster. **G** Change in ISG score for pDC over time in patients treated with PEG-IFN-λ or placebo. **H** Change in ISG score for Intermediate monocytes (*IFNLR1* negative) over time in patients treated with PEG-IFN-λ or placebo. *Y*-axis denotes the change in ISG score compared to Day 0. Change in ISG score for (**I**) B cells 1 and (**J**) Memory B cells when enriched for only cells that have non-zero *IFNLR1* expression. Blue line is the average ISG score for 5 PEG-IFN-λ treated patients, shown individually in solid grey lines. Red line is the average ISG score for 4 placebo treated patients, shown individually in dotted grey lines.

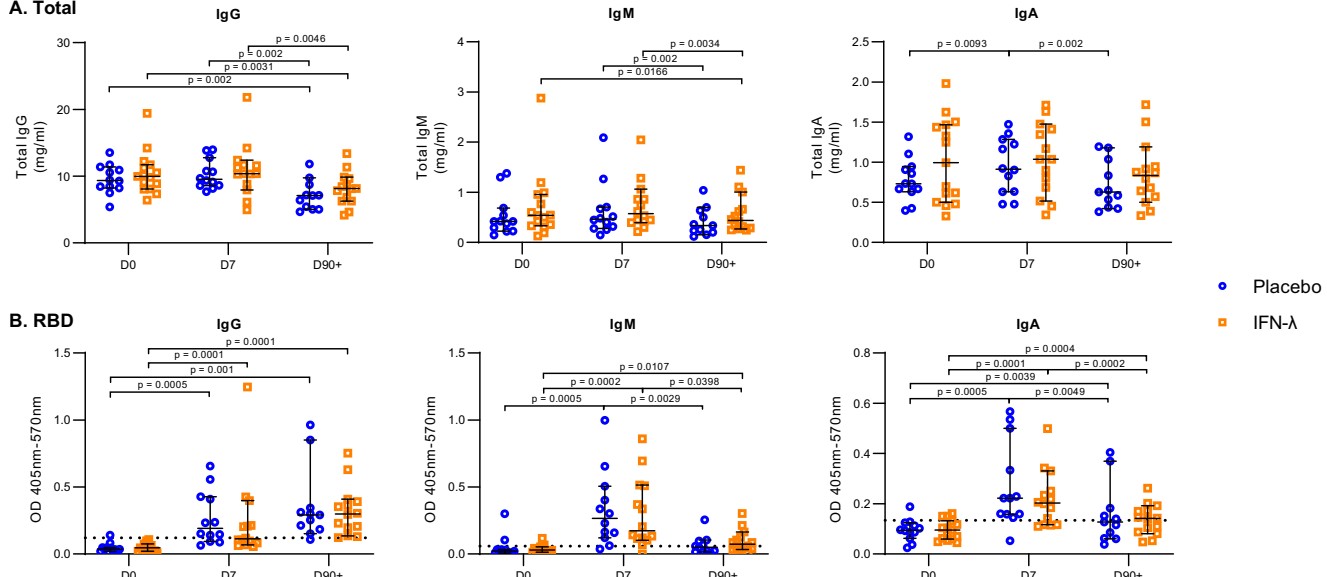

**Fig. 2 | Comparison of total and SARS-CoV-2 spike-RBD plasma antibody levels between placebo and PEG-IFN-λ treated patients at day 0, day 7, and day 90 + post-enrollment. A**, **B** Total IgG, IgM and IgA (**A**) and RBD-specific IgG, IgM and IgA (**B**) in patient plasma were measured by ELISA from samples collected day 0 (D0), day 7 (D7) and day 90 + (D90 + ) post enrollment in the phase II clinical trial. Dashed line in (**B**) represents the mean + 2 SD of results obtained from 8 pre-

pandemic plasma controls collected in 2018–2019. Each dot represents a different patient. *N*-values are as follows: Placebo, D0 (*n* = 12), Placebo, D7 (*n* = 12), Placebo, D90 + (*n* = 11), IFN-λ, D0 (*n* = 15), IFN-λ, D7 (*n* = 14), IFN-λ, D90 + (*n* = 14). *P*-values shown are based on a two-sided Wilcoxon signed-rank test. Bar lines represent median and 95% CI. Source data are provided as a Source Data file.

displayed positive GzmB responses towards M, E, N, and S protein at any of the time points (Supplementary Fig. 2). The number of responses towards E were lower than responses to the other proteins. The median envelope responses across all time points for IFN-γ, IL-2, and polyfunctional responses never exceeded 13 SFUs/million PBMCs (Supplementary Fig. 3). Therefore, we focused our analysis on T cells responsive to the S, N, and M proteins and the effector functions IFN-γ and IL-2.

We observed similar kinetics in the IFN-γ + T cell responses targeting S and N between the two treatment groups, with T cell responses peaking at D7 followed by a significant reduction by D90 + . We did not observe any differences in the magnitude of IFN-γ + T cell responses between placebo- and PEG-IFN-λ-treated patients (Fig. 3A). M-specific IFN-γ responses did not change over time (Fig. 3A). IL-2 + T cell responses followed a similar trend, peaking at D7 and declining by D90 + . In contrast to IFN-γ, M-specific IL-2 responses increased between D0 and D7 in both groups, and the increase between D0 and D90 + was maintained in PEG-IFN-λ-treated patients (Fig. 3B). No significant differences in the magnitude of IL-2 + T cell responses were observed between placebo and PEG-IFN-λ-treated patients. Polyfunctional responses followed the same profile as individual cytokines, peaking at D7 for all SARS-CoV-2 antigens with no significant differences between placebo- and PEG-IFN-λ-treated groups (Fig. 3C). In addition to the lack of differences in the magnitude of T cell responses between patient groups, we did not observe differences in the

proportion of patients with a positive response between placebo- and PEG-IFN-λ-treated patients at the three time points (Supplementary Fig. 1). We also noted no differences in the breadth of responses in the two groups, with both groups showing similar proportions of antigen-specific responses at the three time points. There was no significant difference in the time between symptom onset and enrollment between the patient groups (Supplementary Fig. 4).

Since T cell responses aid in B cell responses, we determined if T cell cytokine data correlated with RBD-specific antibody production. We found significant correlations between the interferon-gamma (IFN-γ), interleukin-2 (IL-2), and polyfunctional (IFN-γ+ & IL-2+) spike-specific T cell responses and RBD-IgG and IgA levels at D90+, but not at D0 or D7 (Supplementary Table 4). RBD-specific IgM antibody levels and spike-specific T cell responses did not significantly correlate at any time point (Supplementary Table 4).

Overall, these results indicate that although COVID-19 patients in our trial mounted T cell responses to multiple SARS-CoV-2 proteins, PEG-IFN-λ treatment had no effect on the magnitude or functionality of virus-specific T cell responses over time.

## SARS-CoV-2-specific T cell responses were delayed in older patients

Having observed that PEG-IFN-λ treatment did not impact the magnitude or kinetics of the T cell response in patients, we investigated

**Table 1 | Patient characteristics for T cell and antibody analyses**

| | Placebo | PEG-IFN-λ | Total |
|---|---|---|---|
| **T cell analysis** | | | |
| **Total #** | 17 | 21 | 38 |
| Day 0 | 17 | 21 | 38 |
| Day 7 | 16 | 17 | 33 |
| Day 90+ | 15 | 19 | 34 |
| **Sex** | | | |
| Female | 7 | 8 | 15 |
| Male | 10 | 13 | 23 |
| **Median age, years (range)** | 41 (22–63) | 47 (21–61) | 45 (21–3) |
| ***IFNL4* genotype** | | | |
| ΔG | 0 | 1 | 1 |
| TT/ΔG | 9 | 9 | 18 |
| TT | 8 | 11 | 19 |
| **Antibody analysis Time Points** | | | |
| Day 0 | 12 | 15 | 27 |
| Day 7 | 12 | 14 | 26 |
| Day 90+ | 11 | 14 | 25 |
| **Sex** | | | |
| Female | 4 | 8 | 12 |
| Male | 8 | 7 | 15 |
| **Median age, years (range)** | 45.5 (22-62) | 42 (21-61) | 43 (21-62) |
| ***IFNL4* genotype** | | | |
| ΔG | 0 | 0 | 0 |
| TT/ΔG | 6 | 7 | 13 |
| TT | 6 | 8 | 14 |

additional demographic variables associated with severe COVID-19 disease. During the course of the pandemic, older COVID-19 patients have been found to be at an increased risk of severe complications and death[24–28]. To determine if these observed outcomes may be attributed to virus-specific T cell responses, we compared SARS-CoV-2-specific T cell responses between patients below and above the median age of the cohort (median age = 45, n = 19 for both groups). No patients were exactly 45 years old. We found that older patients had significantly reduced responses towards S and N proteins at D0. The median IFN-γ SFUs/million PBMCs towards S protein at D0 was 41.6 in older patients, compared to 323.0 in younger patients ($p = 0.0080$, Fig. 4A). The median responses towards the N protein at D0 in older patients was 5.88 and 173.2 in younger patients ($p = 0.0009$, Fig. 4A). Notably, older patients had a similar number of M-specific IFN-γ SFUs as the younger group at D0 ($p = 0.23$, Fig. 4A). Similar trends were seen with IL-2 and polyfunctional SFUs between older and younger patients towards S, N, and M proteins at D0. From D7 onwards, these differences were no longer detected with T cell responses equalized between older and younger patients. However, D90 + M-specific IL-2 and polyfunctional responses were higher in older patients ($p = 0.0348$ and $p = 0.0491$, respectively, Fig. 4B-C). In the trial, five patients (4 placebo and 1 PEG-IFN-λ) required emergency room care or hospitalization, all of whom were above age 45. Despite the delay in T cell responses seen in older patients, PEG-IFN-λ treatment was still able to reduce viral load regardless of their age, with similar responses seen in those above and below 45 (Fig. 5). There were no differences in time from symptom onset to enrollment or baseline viral load, between age groups (Supplementary Fig. 5). The impact of age was specific to the T cell compartment as no significant differences in total or RBD-specific IgG, IgM or IgA levels were observed between those younger or >45 years

(Supplementary Fig. 6). These data suggest that the antiviral activity of IFN-λ acts independently of the cellular immune response.

We also assessed patient characteristics such as sex and the *IFNL4* genotype, where both have been associated with more severe COVID-19 outcomes. Male patients have been more likely to suffer worse COVID-19 outcomes, as do those with the ΔG *IFNL4* genotype[29–33]. The same *IFNL4* allele has previously been noted to affect spontaneous and IFN treatment-driven clearance of hepatitis C virus (HCV)[34,35]. No differences in SARS-CoV-2-specific T cell responses were found by sex or *IFNL4* genotype (Supplementary Figs. 7 and 8). Although there were differences in antibody levels by sex and *IFNL4* genotype, a clear pattern was not observed (Supplementary Figs. 9 and 10). Our data suggest that age, but not sex or *IFNL4* genotype, negatively impacts development of the SARS-CoV-2-specific T cell response.

**Older COVID-19 patients have less diverse IFN-γ T cell responses to SARS-CoV-2 in early infection**

To better understand the impact of age on the delayed T cell response, we aggregated responses to SARS-CoV-2 proteins for each patient and arranged patients based on age (left to right on graphs) for each time point tested. For IFN-γ responses at D0, younger patients displayed a greater diversity in their T cell repertoire, targeting all three SARS-CoV-2 proteins whereas older patient responses were largely directed towards the membrane protein. Quantitatively, 16/19 (84.2%) of older patients attributed more than half of their total IFN-γ responses to membrane protein alone at D0. Meanwhile, only 6/19 (31.6%) of younger patients shared this result ($p = 0.0031$, Fig. 6A). By D7, T cell diversity expanded in older patients and only 9/15 (60%) patients displayed a dominant M response, which was not significantly different from younger patients (5/18 (27.8%); $p = 0.1307$, Fig. 6A). IFN-γ response to S and N contracted in all patients by D90 + and the overall response was dominated by M at this timepoint.

IL-2 responses showed a different antigen-specific distribution and even greater differences in magnitude based on age. The majority of IL-2+ responses were directed towards S and N and the magnitude of IL-2 responses was clearly higher in younger patients at D0 (Fig. 6B). Similar to IFN-γ, at D7 IL-2 + T cells became detectable in the older patients, also primarily targeting the S and N proteins. However, unlike IFN-γ, IL-2+ responses at D90 + remained distributed between S and N, with M-specific T cells contributing less to the overall IL-2 response (Fig. 6B). The polyfunctional T cell response followed the same pattern as observed for IL-2 (Fig. 6C). Together our findings show that the early IFN-γ + antigen-specific T cell repertoire differed significantly by age and that early IL-2 responses, critical for T cell function, were significantly reduced in magnitude in older patients.

**Discussion**

Our Phase II clinical trial data demonstrated that a single subcutaneous injection of PEG-IFN-λ (180 μg) showed efficacy as an early antiviral treatment for COVID-19. Here, we show that specific immune cells in the peripheral blood were responsive to PEG-IFN-λ, but this responsiveness did not modulate peripheral adaptive immunity to SARS-CoV-2, either positively or negatively. However, early sampling revealed that older patients displayed a delayed T cell response towards SARS-CoV-2, showing a less diverse and less functional early response. Overall, our findings show that accelerated clearance of SARS-CoV-2 by PEG-IFN-λ was mediated by induction of the antiviral ISG response without major effects on B and T cell immunity, an advantage in older patients where the T cell immune response was delayed.

Our results revealed that subsets of immune cells within PBMCs from COVID-19 patients expressed *IFNLR1* and were able to respond to PEG-IFN-λ treatment by upregulating ISGs. To our knowledge, this is the first report demonstrating specific human immune cell subsets are sensitive to PEG-IFN-λ treatment in vivo. PEG-IFN-λ was not given to healthy individuals. Therefore, we were not able to determine if the ISG

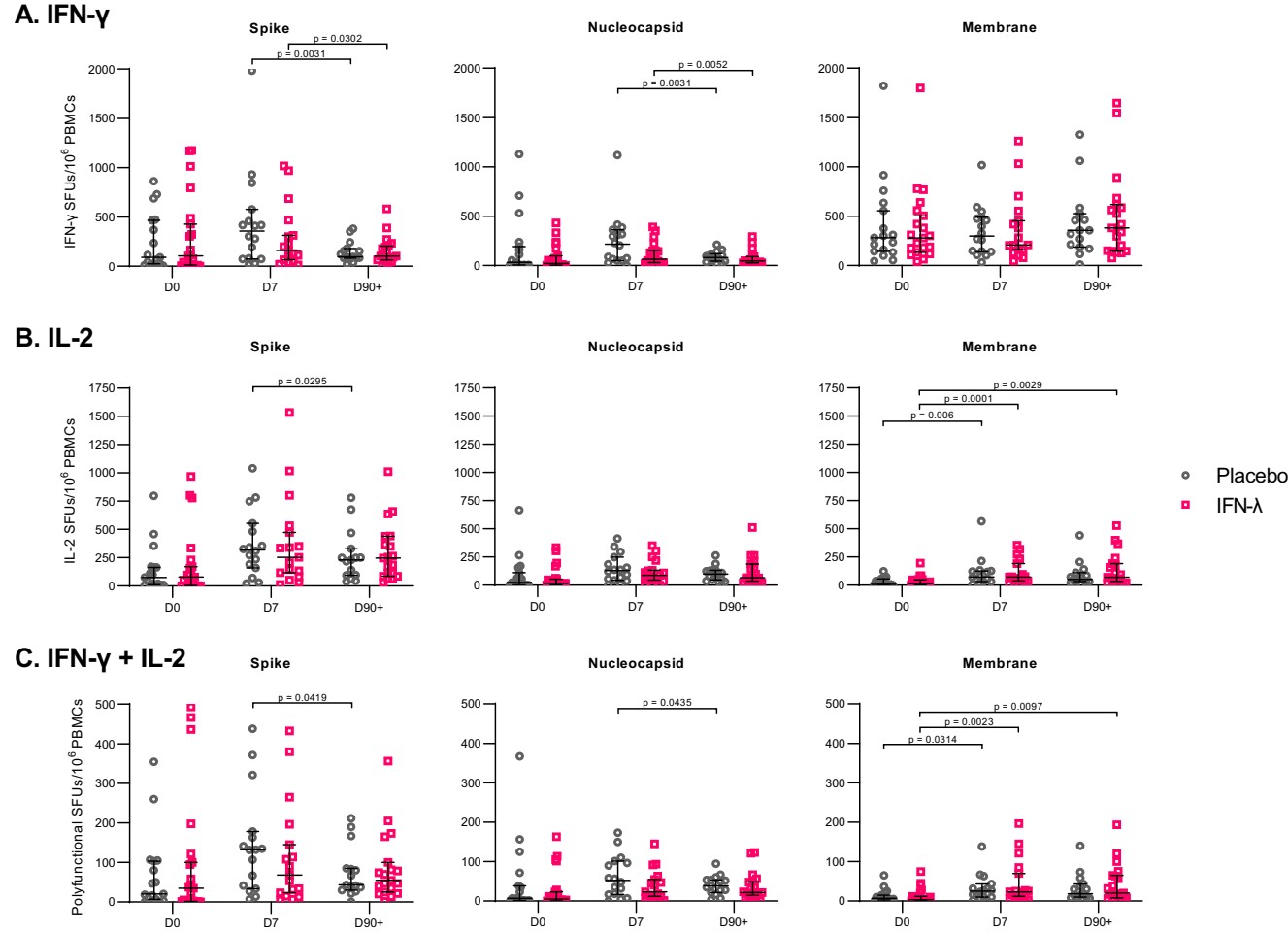

**Fig. 3 | Comparison of T cell responses between placebo and PEG-IFN-λ treated COVID-19 patients at day 0, day 7, and day 90 + post enrollment. A** IFN-γ **B** IL-2 **C** Polyfunctional (IFN-γ + IL-2) T cell responses (as SFUs per 10⁶ PBMCs) against structural SARS-CoV-2 protein peptide pools were quantified ex vivo using FluoroSpot assays. Each dot represents a different patient. *N*-values are as follows:

Placebo, D0 (*n* = 17), Placebo, D7 (*n* = 16), Placebo, D90 + (*n* = 15), IFN-λ, D0 (*n* = 21), IFN-λ, D7 (*n* = 17), IFN-λ, D90 + (*n* = 19). *P*-values showed are based on a two-sided Wilcoxon signed-rank test.. Bar lines represent median and 95% CI. Source data are provided as a Source Data file.

response was dampened by SARS-CoV-2 proteins implicated from in vitro studies but the clinical data show clear benefit of IFN-λ therapy[8,36,37]. Our in vivo results agree with earlier healthy donor in vitro studies demonstrating IFN-λ responsiveness, with purified pDCs and B cells as the top responders and monocytes and natural killer cells as non-responders[20,23]. Subsets of CD8 + T cells, despite expressing *IFNLR1*, did not respond to PEG-IFN-λ in vivo. This is in contrast to our previous results where in vitro IFN-λ3 treatment of healthy donor CD8 + T cells led to an upregulation of antiviral ISGs measured by reverse transcription quantitative real-time PCR (RT-qPCR)[20]. Deeper analysis of the *IFNLR1* + CD8 T cells revealed relatively low expression levels for both receptor chains, which may induce a response that falls below the sensitivity of scRNAseq. This would be in line with our in vitro studies, where IFN-λ induced ~10-100 fold greater ISG fold changes in primary bronchial lung epithelial cells compared to purified B cells[20], confirming the potent nature of IFN-λs at the site of SARS-CoV-2 infection. We also noted high baseline ISG expression in all patients, indicating the presence of endogenous IFNs. The endogenous IFNs were likely type I IFN, which would explain the variability in ISG module scores seen in monocytes from placebo control patients enrolled at different days after symptom onset. This was anticipated due to the acute virus infection. Despite IFN-λ being exempt from desensitization typical of IFN-α repeat responses[38], the high expression

of ISGs likely limited the magnitude of ISG induction upon PEG-IFN-λ treatment in immune cells. To gain further insight into the immuno-modulatory properties of IFN-λ therapy, we also considered investigating differences between responders and non-responders in the IFN-λ treatment arm. However, only one IFN-λ treated patient was classified as a non-responder in the clinical study, limiting our ability to draw any conclusions related to non-response. Overall, we demonstrate that immune cells respond to IFN-λ in vivo despite an ongoing SARS-CoV-2 infection. However, viral clearance was likely driven by direct ISG activation, as demonstrated in our recently published paper measuring *OAS1* activity, rather than promotion of adaptive immune responses[39].

Similar to previous reports, we detected virus-specific T cell responses in acute and convalescent COVID-19 patients targeting SARS-CoV-2 structural proteins, including spike, nucleocapsid, and membrane[40–45]. Spike- and nucleocapsid-specific T cells dominated the T cell response, peaking at D7 and displaying the broadest functionality. Membrane-specific T cells displayed different kinetics for IFN-γ, with detectable responses at D0 that did not change significantly over time. PEG-IFN-λ treatment had no impact on the kinetics, magnitude, functionality, or maintenance of a functional memory T cell pool compared to placebo. However, age, a key variable associated with severe COVID-19 disease outcomes, impacted the generation of SARS-CoV-2-specific T cell immunity[28,46–48]. Both IFN-γ and IL-2 responses

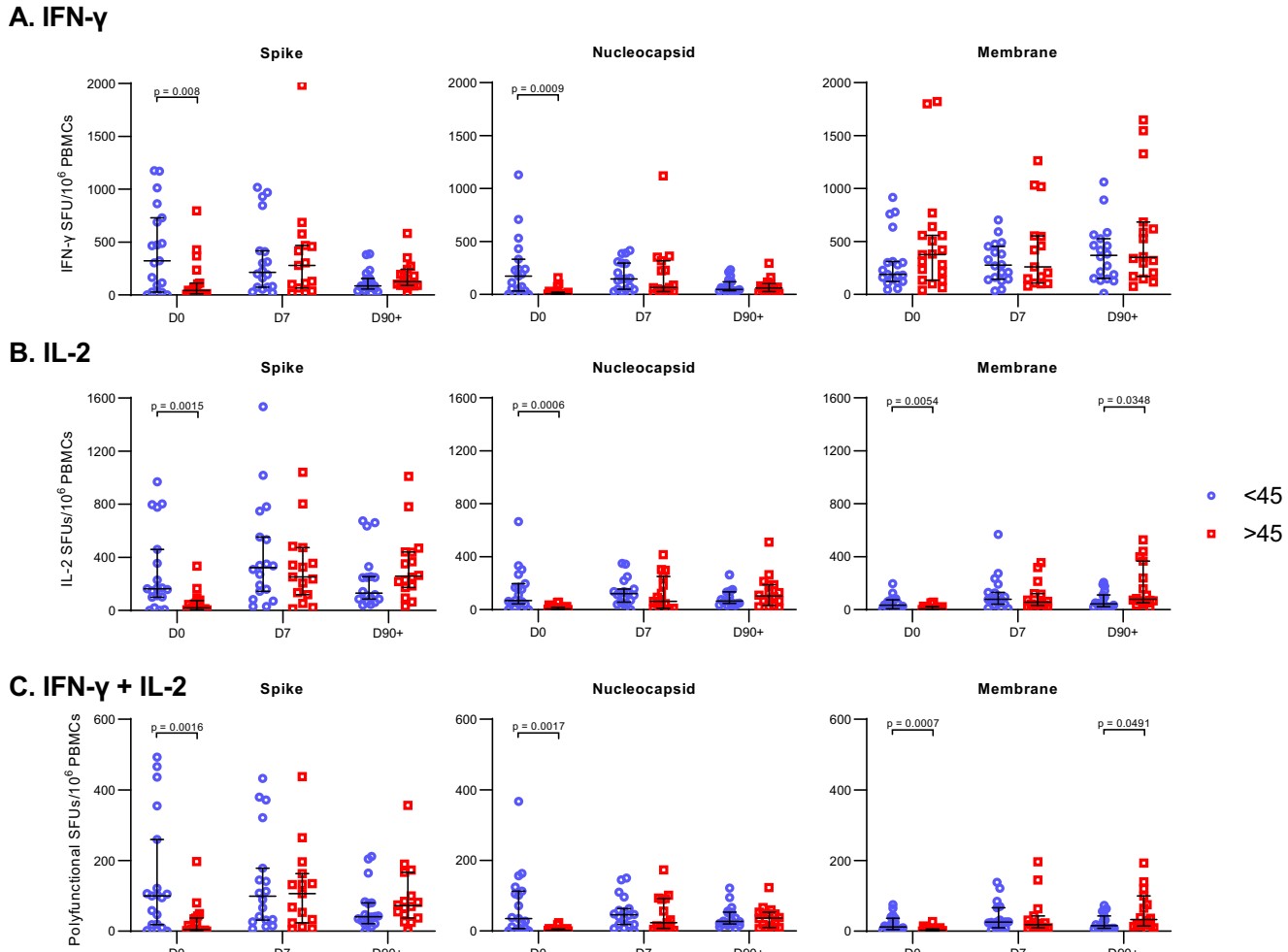

**Fig. 4 | Differences in T cell responses between patients below and above the median age at day 0, day 7, and day 90 + post-enrollment. A** IFN-γ **B** IL-2 **C** Polyfunctional (IFN-γ + IL-2) T cell responses (as SFUs per 10⁶ PBMCs) against structural SARS-CoV-2 protein peptide pools were compared between patients above and below the median age of the whole cohort (45 years old). Each dot represents a different patient. Blue circles are <45 y and red squares are >45 years. *N*-values are as follows: <45, D0 (*n* = 19), <45, D7 (*n* = 18), <45, D90 + (*n* = 18), >45, D0 (*n* = 19), >45, D7 (*n* = 15), >45, D90 + (*n* = 16). *P*-values showed are based on a two-sided Wilcoxon signed-rank test. Bar lines represent median and 95% CI. Source data are provided as a Source Data file.

were significantly delayed in patients over 45 years old. Notably all five patients in the trial who required hospitalization were over this age threshold. Our data are consistent with a recent report showing impaired naïve CD8 + T cell priming in older patients[49,50]. Other studies have also assessed age-related differences in T cell responses, observing decreased cytotoxic CD8 + responses, lower IFN-γ/higher IL-2 secreting CD4 + T cells, and uncoordinated adaptive immune responses in older patients[51–53]. In assessing the interaction between treatment and age groups, we found viral load decline was similar in younger and older groups when stratified by treatment, suggesting that PEG-IFN-λ treatment can have an antiviral effect despite the differences in T cell responses. In one fatal COVID-19 case, the patient did not exhibit detectable SARS-CoV-2-specific CD4 + or CD8 + responses after 2 weeks post-symptom onset, highlighting the importance of mounting T cell responses during early infection[53]. However, some of these studies either had a lower number of participants or recruited participants during a wide range of time after symptom onset. A strength of our study was baseline samples were collected within 7 days of symptom onset in patients with a laboratory-confirmed diagnosis, providing better insight into the immune response early in the infection. Altogether, our findings show delayed T cell responses early after infection in older individuals, potentially exposing them to

greater severity of outcomes, which may be compensated by early therapeutic intervention with PEG-IFN-λ.

Given our previous study showed in vitro exposure to IFN-λ negatively impacted influenza vaccine antibody responses[22], we anticipated a negative impact on antibody production. However, despite detecting B cells were responsive to PEG-IFN-λ in peripheral blood, no difference in the levels of RBD-specific IgM, IgA, or IgG were measured in patient plasma between placebo and PEG-IFN-λ groups. This indicates that one dose of PEG-IFN-λ was not sufficient to alter systemic antibody levels. RBD-specific IgG antibodies were still elevated above baseline in most patients at D90 +, indicating long-term circulating levels in plasma. Unlike T cell responses, there was no significant impact of age on antibody levels when we compared those above or below the median of age 45. Age has been negatively correlated with SARS-CoV-2 antibody levels, although greatest differences have been documented in those over 60[54,55] and we only had 2 participants enrolled over this age. Whether multiple injections of PEG-IFN-λ could impact B cell function or responses by age, or whether memory B cell persistence and function at mucosal sites were altered requires further investigation.

We also assessed sex and *IFNL4* genotype during our analysis since both have been associated with COVID-19 disease outcomes[29–33].

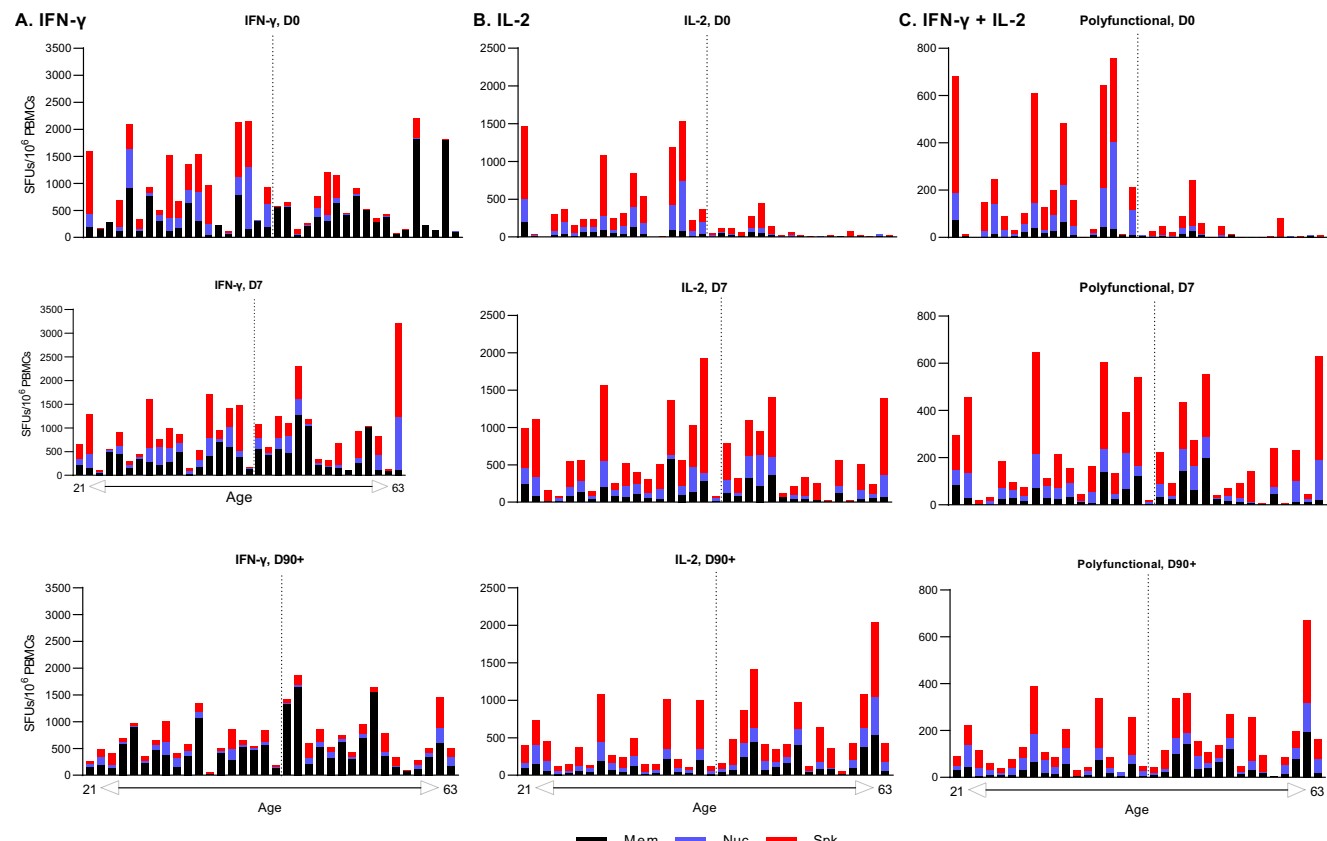

**Fig. 5 | Viral load decline after placebo or PEG-IFN-λ treatment stratified by age.** Mean viral load decline from baseline is shown as quantified for the Phase II Clinical trial by RT-qPCR per day after trial enrollment. No significant differences were observed between the age groups within their respective treatment groups (Wilcoxon rank sum test). Error bars represent the standard error. Source data are provided as a Source Data file.

**Fig. 6 | T cell responses at day 0, day 7, and day 90 + in increasing order of age.** **A** IFN-γ **B** IL-2 **C** Polyfunctional (IFN-γ + IL-2) T cell responses (as SFUs per 10⁶ PBMCs) against structural SARS-CoV-2 protein peptide. The dashed line represents the median age of 45 years. N-values are as follows: D0 (n = 38), D7 (n = 33), D90 + (n = 34). Mem membrane (black), Nuc nucleocapsid (blue), Spk spike (red). Source data are provided as a Source Data file.

Multiple groups have found a greater proportion of male patients suffer from more severe COVID-19 outcomes[29–31]. While we did not find sex differences in T cell responses in our patient cohort, we were able to see some significant differences in antibody levels (i.e., Total IgM and RBD-specific IgA) but the relevance of these differences remains unclear. In chronic HCV infections, *IFNL4* genotype has been found to negatively affect the efficacy of PEG-IFN-α treatment and the probability of spontaneous viral clearance in those with the ΔG rs368234815 genotype[34,35,56]. Similar to our findings, a recent study found no associations with SARS-CoV-2-specific CD8 + T cell responses or antibody levels and *IFNL4* genotype[57]. However, *IFNL4* variants have been found to be associated with the severity of and predisposition to acquiring COVID-19[32,33], suggesting that *IFNL4* genotype may affect innate

immune responses, rather than adaptive responses. Of the five patients in our cohort who required hospital care, four had the risk-associated genotype at rs368234815. Additional analyses of the effects of sex and *IFNL4* genotype on SARS-CoV-2-specific T cell and antibody responses would be useful, as our findings are limited by small sample size.

To summarize, our analyses demonstrate that a single dose of PEG-IFN-λ accelerates SARS-CoV-2 clearance without affecting virus-specific T cell responses or antibody production in mild-to-moderate acute SARS-CoV-2 infection. Compared to current antiviral treatments for COVID-19, PEG-IFN-λ treatment is broad-acting, effective with a single dose, and is less likely to be affected by new variants or resistance mutations. This supports future use of PEG-IFN-λ as an early treatment option because it provides beneficial antiviral effects

without negative consequences on adaptive immunity. This aspect may be particularly relevant for older COVID-19 patients who may have naturally delayed T cell responses to SARS-CoV-2 early in an infection.

## Methods

### Ethical statement and human subjects

Subjects were recruited for a randomised, double-blind, placebo-controlled study from outpatient testing centers at six institutions in Toronto, Canada. Eligible individuals had a SARS-CoV-2 infection confirmed by nasopharyngeal swab and were enrolled within 7 days of symptom onset or first positive test if asymptomatic. The health research ethics boards of all participating institutions approved this study (University of Toronto, University of Alberta, University of Manitoba) using samples collected from the approved trial, which was registered (NCT04354259) and done under a Clinical Trial Application approved by Health Canada. All participants provided written informed consent. Participants were compensated for their time ($50 CAD) and any travel expenses at each study visit. Additional trial information is detailed in Feld et al. (2021)[11].

### Plasma collection and PBMC isolation

Freshly collected blood samples in acid citrate dextrose (ACD) tubes were centrifuged and plasma was frozen and stored in −80 °C. Remaining whole blood was used for PBMC isolation, using SepMate PBMC Isolation Tubes and the Lymphoprep density gradient medium (STEMCELL Technologies). Isolation was conducted according to the manufacturer's instructions. PBMCs were subsequently frozen at −80 °C overnight and stored in liquid nitrogen for long-term storage.

### PBMC scRNAseq

PBMCs were thawed quickly at 37 °C and washed to remove freezing media. Pelleted cells were resuspended in PBS + 0.1% low endotoxin BSA and counted using 0.4% Trypan blue solution (Thermo Fisher Scientific). Samples had average of 87% viability after thawing. Cells were resuspended in PBS + 0.1% low endotoxin BSA in an appropriate volume to achieve a concentration of 1000 cells/ml. This cell suspension was used to generate the gel-beads + cell emulsion by the 10X Chromium Controller (PN-1000202) using the Chromium Next GEM Single Cell 5′ v2, Chromium Next GEM Chip K Single Cell Kit and Dual Index Kit TT Set A. Reverse transcription, cDNA amplification, library preparation, and sample barcoding were performed following the available manufacturer's protocol. Finally, sample libraries were pooled and sequenced in Illumina HiSeq P150 (Sequencing type: Paired-end, single indexing) to an average depth of ~ 35,261 reads per cell (Novogene).

### Preprocessing and analysis of scRNAseq data

FASTQ files were inputted to 10X Genomics Cell Ranger 6.0.0 count tool with default parameters and mapped to the human reference GRCh38-2020 also provided by 10X[58]. Afterwards, the filtered matrix files output by Cell Ranger count tool was used with the aggr tool in Cell Ranger with default parameters. Aggregated expression matrix for all cells was analyzed with the Seurat v4.0.4 R package[59]. Briefly, genes which were expressed in <100 cells were discarded. Low quality cells with <500 genes or cells with high (≥15%) mitochondrial content were filtered out. Furthermore, doublet cells were identified through scDblFinder v1.7.4[60]. Standard Seurat pipeline was run with 5000 variable features retained and 50 principal components (PCs) used for downstream analysis. In addition, Harmony v0.1.0 was used for batch effect correction due to sequencing batch and sex[61]. Number of neighbors for both UMAP projection and SNN network was set at 50. Finally, 0.8 was set as the resolution for the Louvain neighborhood detection. After this first-pass analysis, all clusters which had >50% of cells classified as doublets were discarded along with cells that were classified as doublets. The filtered dataset was run through Seurat again

with the same parameters as previously set. Differentially expressed genes were identified using "FindMarkers" from Seurat with parameters (latent.vars = c("Batch","Sex","nFeature_RNA"), test.use = "LR").

### ISG scoring and visualization

ISG score for each cell was calculated according to Seurat function "AddModuleScore" with default parameters. The list of ISGs which were used to compute the score are in Supplementary Table 1. ISGs for the module were selected based on detection of changes in expression within the dataset between baseline and day 3 and those known to be induced by IFN-λ in our and other's previous studies[20,62–67]. Due to the variability of the scores between patients and by baseline viral load, we calculated the change of ISG score compared to D0. Panel generation and compilation was done through R packages (ggplot2 v3.3.5, ggrepel v0.9.1, patchwork v1.1.1, dplyr v1.0.7, reshape2 v1.4.4)[68–72].

### Total and SARS-CoV-2-specific antibody ELISAs

All plasma was heat-inactivated at 56 °C for 45 min before diluting for ELISA. Total IgG and IgM in plasma were measured via an in-house ELISA with standards and antibodies from Jackson Immunoresearch (Donkey anti-human IgG Fcgamma specific (#709-005-098), Chrom-Pure Human IgG (#009-000-003), Goat anti-human IgG (Fcg)-Alkaline phosphatase conjugated (#109-055-190), Donkey anti-human IgM (#709-005-073), ChromPure Human IgM (#009-000-012), Goat anti-human IgM (Fc5u)-Alkaline phosphatase conjugated (#109-055-129)). Total IgA measurements were quantified using an ELISA kit from STEMCELL Technologies. Our SARS-CoV-2 RBD-ELISA was optimized based on a study by Amanat et al. (2020)[73] using purified spike RBD (wild-type) supplied as a gift from the lab of Dr. Michael Houghton (University of Alberta). 96-well plates (Corning 96-well EIA/RIA Easy Wash™) were coated overnight at 4 °C with 50 ul RBD (1.5 ug/ml) in PBS. After washing, wells were blocked with 3% non-fat milk in PBS for 2 h at room temperature. Optimal dilutions were determined balancing background and detection limits where the final dilutions chosen were 1:100 for IgG, 1:40 for IgM and 1:40 for IgA. Plasma was diluted in 1% non-fat milk in PBS. Secondary antibodies (goat anti-human Ig alkaline phosphatase) were from Jackson Immunoresearch (anti-human IgG, IgM- same as above) or Thermo Fisher Scientific (Goat anti-human IgA (α chain)- Alkaline phosphatase conjugated (#A18784)) and PNPP substrate was from Thermo Fisher Scientific. 8 pre-pandemic plasma samples collected in 2018 or earlier were used to determine a baseline background (shown as dotted line on graphs). One positive control was run on each plate to normalize readings between plates.

### Peptide pools

12- to 15-mer peptides overlapping with 10 amino acids residues spanning the full sequences of the wild-type SARS-CoV-2 membrane (M), envelope (E), nucleocapsid (N), and spike (S) proteins were used to stimulate PBMCs (BEI Resources). Peptides were reconstituted with 20 uL of DMSO (50 mg/mL) and pooled to form 8 peptide pools (Supplementary Table 5). A peptide pool made up of epitopes from cytomegalovirus, Epstein-Barr virus, and influenza virus (CEF) was used as a positive control and contained 32 peptides.

### PBMC stimulation and FluoroSpot

PBMCs were thawed at 37 °C, washed with Hanks' Balanced Salt Solution and resuspended in AIM V treated with primocin (Life Technologies). 300,000 PBMCs were added in duplicates to each well on a 96-well FluoroSpot plate and treated with peptide pools at a concentration of 5 ug/mL for 24 h at 37 °C. In some instances, 200,000 PBMCs were plated. The negative control consisted of 0.46% DMSO, which is the highest concentration of DMSO cells in the peptide treated wells were exposed to. Positive controls consisted of anti-CD3/CD28 beads

(Dynabeads; Thermo Fisher Scientific) and a CEF peptide pool (Gen-Script, 5 ug/mL). T cell responses were measured using a 3-colour FluoroSpot assay (Cellular Technology Limited (CTL) ImmunoSpot) measuring human IFN-γ (green), IL-2 (yellow), and GzmB (red) secretion. Assays were conducted according to the manufacturer's instructions. Plates were scanned using the CTL ImmunoSpot S6 Analyzer. Spot forming units (SFUs) were counted using ImmunoSpot Software. All SFU counts were normalized by subtracting the background DMSO-stimulated SFU count of the individual patient time point.

## Viral load

Quantitative SARS-CoV-2 viral titer results were analyzed using patient mid-turbinate (MT) swabs. 400uL of the MT swab viral transport media was taken and viral genetic material was extracted using the NucliSENS EasyMAG platform (bioMerieux, Marcy-l'Etoile, France). Afterwards, a one-step reverse transcriptase quantitative PCR was performed on Rotorgene Q (Qiagen, Hilden, Germany) using primers and probes for the SARS-CoV-2 E gene. Standard curves were generated using dilutions of synthetic plasmids containing segments of the E gene (GenScript, USA). The number of viral copies per uL was measured based on the dilutions of the plasmid, with interpolation of the sample values done using GraphPad Prism. Samples with no Ct value or a value below the level of quantification were considered to be negative. Those below the limit of quantification (~20 copies/mL) were arbitrarily assigned a value of 10 copies/mL. Additional methods information is detailed in Feld et al. (2021)[11]. Viral load decline was compared between groups above and below age 45 with and without PEG-IFN-λ treatment, after controlling for baseline viral load.

## *IFNL4* genotyping

The interferon lambda-4 genotype (*IFNL4*) was assessed by sequencing rs368234815 in genomic DNA from whole blood. Briefly, the genomic DNA was extracted from whole blood using QIAamp DNA Blood Kit (Qiagen, Canada) according to the manufacturer's instructions. The extracted DNA was amplified using 5'- GCACTGCAGACAGGAGTGAG -3' and 5'- TCGTAGCGGTCCCTCAG-3' as forward and reverse primers, respectively. Purified PCR amplicons were directly sequenced using 5'-GACGTCTCTCGCCTGCT-3' as a sequencing primer by Sanger method to determine the genotype which was categorized as TT or non-TT (ΔG/T or ΔG/ΔG)[34,35,56].

## Statistical analysis

Data between time points were compared using paired Wilcoxon *t*-tests with patients missing measurements for time points excluded from analysis. Data between patient treatment groups (placebo versus PEG-IFN-λ) were compared using two-sided Mann–Whitney *U*-tests. Correlation analysis was conducted using non-parametric, Spearman rank correlation tests. Chi-square tests with Yates' correction were conducted for comparing the proportion of positive responses between treatment groups and the proportion of individuals with high membrane responses between age groups. Viral load testing was compared at each time point, correcting for multiple comparisons (Bonferroni) and a logistic regression model assessed the association of treatment with viral load decline controlling for baseline viral load and age above or below 45. Statistical analysis was conducted on GraphPad Prism version 9.3 and R version 4.1.1. Significant differences are labelled as $*p < 0.05$; $**p < 0.01$; $***p < 0.001$; $****p < 0.0001$. Insignificant differences remain unlabelled.

## Reporting summary

Further information on research design is available in the Nature Portfolio Reporting Summary linked to this article.

## Data availability

The single-cell RNA-sequencing data generated in this study have been deposited in the Gene Expression Omnibus (GEO) database under accession code GSE215814. Source data are provided with this paper.

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

## Acknowledgements

We thank Joaquín López-Orozco and Karyn Berry-Wynne from the University of Alberta High Content Analysis core facility and Tyrrell lab, respectively, for their assistance with preparing the 10x genomic libraries. We thank Julia Casey, Conan Chua, and the clinical team at the Toronto Centre for Liver Disease for their assistance with sample collection, preparation, and method support. Thank you to Dr. Michael Houghton for the kind gift of recombinant SARS-CoV-2 spike RBD protein. Toronto COVID-19 Action Initiative – University of Toronto (FELD_TCAI_2020; J.J.F.). CIHR Operating Grant (COVID-19 Rapid Research Funding Opportunity) (D.L.J.T., J.J.F.). University of Toronto (J.J.F.). Ontario First COVID-19 Rapid Research Fund (J.J.F.). Toronto General and Western Hospital Foundation (J.J.F.). University of Manitoba Start-Up Funding (D.M.S.).

## Author contributions

Conceptualization: D.M.S., D.L.J.T., A.J.G., J.J.F. Data Curation: D.M.S., D.L., Y.G., M.A.Z., D.P. Formal Analysis: D.M.S., D.L., Y.G., M.A.Z., D.P., B.E.H. Funding acquisition: D.M.S., D.L.J.T., J.J.F. Investigation: D.M.S., D.L., Y.G., M.A.Z., D.P. Methodology: D.M.S., D.L., Y.G., M.A.Z., D.P., B.E.H., A.J.G., J.J.F. Project administration: D.M.S., J.J.F., A.J.G. Resources: D.M.S., D.L.J.T., J.J.F., A.J.G. Software: Y.G. Supervision: D.M.S., J.J.F., A.J.G.. Validation: D.M.S., D.L., Y.G., A.J.G., J.J.F. Visualization: D.M.S., D.L., Y.G., D.P. Writing—original draft: D.M.S., D.L., Y.G., A.J.G. Writing—review and editing: D.M.S., D.L., Y.G., M.A.Z., D.P., B.E.H., D.L.J.T., J.J.F., A.J.G. J.J.F. and A.J.G. contributed equally as senior authors.

## Competing interests

Peginterferon lambda was provided to University Health Network (JJF) by Eiger Biopharmaceuticals for the conduct of the study. The authors declare no competing interests.
