## [Peer Review File · Nature Communications]

Interferon- λ treatment accelerates SARS-CoV-2 clearance despite age-related delays in the induction of T cell immunityREVIEWER COMMENTS

Reviewer #1 (Remarks to the Author):

Summary

In "IFN- λ treatment accelerates SARS-CoV-2 clearance despite age-related delays in the induction of T cell immunity" Santer et al. attempt to decipher the mechanism(s) that underlie their previous finding in a double-blind, placebo-controlled trial that PEG-IFN- λ has anti-viral properties (Feld, Kandel et al. 2021). In that study they showed that treatment with PEG-IFN- λ significantly reduced SARS-CoV-2 viral load by 2.42 log₁₀ copies per mL at day 7 with 80% vs. 63% of subjects having undetectable virus in the peg-interferon group and placebo groups, respectively (Feld, Kandel et al. 2021). The authors hypothesized that treatment with PEG-IFN- λ might have induced a more robust SARS-CoV-2 specific T cell and/or antibody response. The data is largely negative in that PEG-IFN- λ did not induce either a stronger T cell or antibody response. They do not describe the mechanism(s) that IFN- λ uses to have an antiviral effect but instead state that PEG-IFN- λ has an antiviral effect regardless of sex and describe an interesting finding that IFN- λ still has an anti-viral effect in adults >45 years of age despite a reduced and delayed T cell response. The authors should show the viral load by time data stratified by treatment group and age to support this statement. The authors also describe the IFNL4 genotype in their subset of 9 subjects, but there is not a clear connection to their original hypothesis and the authors acknowledge the small sample size. Overall, the findings are of general interest to biopharmaceutical companies, regulators and clinicians that are developing anti-viral therapies for SARS-CoV-2 and other viruses and for basic researchers working to better understand the mechanism(s) of action of IFN- λ . Overall, I found the paper difficult to follow as it is a mix of mechanistic studies and COVID-19 epidemiology comparing age, sex and genetics. The T cell and antibody data is negative and the authors then pivot to showing a collection of interesting observational findings from the study unrelated to the PEG-IFN- λ intervention. Towards making the key finding clearer to the reader a proposed title is "IFN- λ treatment accelerates viral clearance despite not inducing a more robust T cell or antibody response to SARS-CoV-2". This title more accurately summarizes the key findings of the paper.

Specific Points

B cells, CD8 T cells and pDCs are responsive to PEG-IFN- λ - Figure 1

The group previously reported that functional type III IFN receptors are expressed on human Immune cells, including B and T cell populations (Santer, Minty et al. 2020) and they use single cell RNAseq analysis to show that IFN- λ R mRNA is expressed in CD8 Granzyme B cells, memory B cells and plasmacytoid dendritic cells in 5 subjects that had PEG-IFN- λ and 4 placebo controls. For someone not familiar with single RNA seq the data in Figure 1 is confusing. It would be helpful if the numbers in panel A were transcribed to panels B and C and define what UMAP_1 vs. UMAP_2 is. It appears the authors used a software program to organize the data in panel D and the cell types are listed 0, 1 5, 2, 7, 11... This is very confusing for the reader and it took a long time to connect the numbers in A to the unlabeled pictures in B and C and then to the unordered numbers in D. In the end, the rather simple summary is that B cells, memory B cells, GzB CD8's and pDCs all express IFN- λ receptor RNA as previously reported.

The authors should note whether the maps shown in A, B and C are unique to COVID-19 subjects or whether they are similar to what would be observed in PBMCs from healthy controls or someone with another viral infection (e.g. influenza or HCV).

The last sentence of this section states "Overall, these analysis demonstrated that IFNLR1+ immune cells in the peripheral blood responded to PEG-IFN- λ treatment in COVID-19 patients." It is not clear, if this is a novel or expected finding. They acknowledge that each of these cell populations were previously demonstrated to respond to IFN- λ in vitro (Yin, Dai et al. 2012, Santer, Minty et al. 2020). The authors should put these current findings in context of what PEG-IFN- λ treatment does to PBMCs in a healthy or virus-infected individual. Specifically, is there something unique about treating a SARS-CoV-2 infection versus another viral infection with PEG-IFN- λ .

Recommendations

- Add labels to B and C
 - Since clusters 2, 6, 10 and 16 (GzB CD8's, B's and pDC's) are key to the authors' hypothesis those should be labeled in B and C.
 - Organize panel D on the X axis from cell group 0 to 20 rather than by genes on the Y axis.
 - Highlight the 4 key IFN- λ R positive cells of interest.
 - Label Panels G – J by cell type instead of cluster type.. E.g. instead of Cluster 16 label the graph pDC.
 - Why show cluster 8 monocytes and not the IFN- λ R positive cluster 2, GzB CD8 cells?
- Ideally, the authors would also have had SC RNAseq on bronchial epithelial cells (recognize

the difficulties in collecting such samples), but they should add text in the discussion of what the ISG

response would look like in PEG-IFN- λ treated epithelial cells.

Antibody levels and antibody responses to SAR-CoV-2 - Figure 2

Y axis for the IgM and IgA data should be changed from IgG to IgM and IgA respectively.

The summary sentence should state that "...and PEG-IFN- λ treatment did not inhibit or increase B cell

antibody responses measured in plasma.

PEG-IFN- λ treatment did not affect T cell responses – Figure 3

It would be helpful to have the FluoroSpot data presented on a log scale (assign 0 values to 0.1) and

state the percentage of subjects without a discernable T cell response. It appears many of the samples

have a value of 0.

SARS-CoV-2 specific T cell responses were delayed in older patients – Figure 4 and 5

Figure 5. add legend describing the different colors in the bar graph (M, spike etc.)

Line 215: The Acceleration of viral decline with PEG-IFN- λ compared to placebo was not affected by age

(OR 0.97, 95% CI 0.93-1.02). This statement is the key foundation to the second half of the paper and

the authors should show a graph of viral load versus time stratified by PEG-IFN- λ vs.

placebo and <45 vs

>45 in the main body of the paper.

Clarify which group includes subjects 45 years of age. (\leq or \geq instead of < or >).

Discussion

Line 327. "To summarize, our analysis demonstrates that a single dose with peginterferon accelerates

viral clearance without affecting virus specific T cell responses or antibody production in mild to

moderate acute SARS CoV-2 infection". The authors don't show this data and they should add a graph(s)

showing VL overtime stratified by age and treatment group.

References

Feld, J. J., C. Kandel, M. J. Biondi, R. A. Kozak, M. A. Zahoor, C. Lemieux, S. M. Borgia, A. K. Boggild, J.

Powis, J. McCready, D. H. S. Tan, T. Chan, B. Coburn, D. Kumar, A. Humar, A. Chan, B. O'Neil, S.

Noureldin, J. Booth, R. Hong, D. Smookler, W. Aleyadeh, A. Patel, B. Barber, J. Casey, R. Hiebert, H.

Mistry, I. Choong, C. Hislop, D. M. Santer, D. Lorne Tyrrell, J. S. Glenn, A. J. Gehring, H. L. A. Janssen and

B. E. Hansen (2021). "Peginterferon lambda for the treatment of outpatients with COVID-19: a phase 2,

placebo-controlled randomised trial." *Lancet Respir Med* 9(5): 498-510.

Santer, D. M., G. E. S. Minty, D. P. Golec, J. Lu, J. May, A. Namdar, J. Shah, S. Elahi, D. Proud, M. Joyce, D.

L. Tyrrell and M. Houghton (2020). "Differential expression of interferon-lambda receptor 1 splice

variants determines the magnitude of the antiviral response induced by interferon-lambda 3 in human

immune cells." PLoS Pathog 16(4): e1008515.

Yin, Z., J. Dai, J. Deng, F. Sheikh, M. Natalia, T. Shih, A. Lewis-Antes, S. B. Amrute, U. Garrigues, S. Doyle,

R. P. Donnelly, S. V. Kotenko and P. Fitzgerald-Bocarsly (2012). "Type III IFNs are produced by and

stimulate human plasmacytoid dendritic cells." J Immunol 189(6): 2735-2745.

Reviewer #2 (Remarks to the Author):

Santer and colleagues report a mechanistic follow-up to their clinical study on use of PEG-IFN- λ in COVID-19 patients, which showed faster viral clearance. The authors conclude that the observed clinical effects were likely the result of ISG induction, as demonstrated by RNAseq showing consistent ISG expression in IFN- λ -responsive immune cell subsets of treated but not placebo-treated patients. At the same time, antibody levels or T cell response were not affected, leading to the conclusion that ISG induction drives virus clearance, which is of importance for treatment of patients with delayed T cell responses such as the elderly. The study is well-conducted and well-written and highly relevant, adding mechanistic data to a recent clinical cohort. Some points should be added to the discussion and some methodological questions need to be clarified. I recommend for publication after minor revision. Specific details:

Line 85: „All 5 patients chosen for scRNAseq analysis who received PEG-IFN- λ demonstrated accelerated virus clearance“ – what about those receiving PEG-IFN- λ who did not demonstrate accelerated virus clearance? ISG induction should work relatively universally – what could lead to reduction of observed clinical effects? High basal IFN- λ and saturation if ISG induction? Please discuss.

Line 104: How was this list of 24 ISGs chosen? There are hundreds of ISGs, how was ensured that this group does lead to biased results? Since the mechanistic conclusions of this manuscript hinges on this result, this needs to be clarified.

Line 107: For Cluster 8 (monocytes), the averages show no elevated ISG response for both groups, but the individual donors analyzed vary considerably and one of the placebo donors has the highest, one of the IFN donors has the lowest response. What is the basis of this, are there individual donor-specific differences in IFN receptor expression in these donors?

Line 167: Text refers to Fig 2A, should probably be 3A instead

General: Is there a risk of increased autoimmunity against IFNs when IFN λ is given exogenously? Please discuss.

What about levels of endogenous IFN- λ pre administration of the exogenous IFN- λ ? Given that ISG levels did not further increase in the IFN group but just stayed high consistently, might this contain clues on when IFNs would be ideally administered? Possibly better at a later timepoint, when infection-induced IFNs drop? Please discuss.

Nature Communications reviewer comments to address:

Reviewer 1

Summary

In “IFN- λ treatment accelerates SARS-CoV-2 clearance despite age-related delays in the induction of T cell immunity” Santer et al. attempt to decipher the mechanism(s) that underlie their previous finding in a double-blind, placebo-controlled trial that PEG-IFN- λ has anti-viral properties (Feld, Kandel et al. 2021). In that study they showed that treatment with PEG-IFN- λ significantly reduced SARS-CoV-2 viral load by 2.42 log₁₀ copies per mL at day 7 with 80% vs. 63% of subjects having undetectable virus in the peg-interferon group and placebo groups, respectively (Feld, Kandel et al. 2021).

The authors hypothesized that treatment with PEG-IFN- λ might have induced a more robust SARS-CoV-2 specific T cell and/or antibody response. The data is largely negative in that PEG-IFN- λ did not induce either a stronger T cell or antibody response. They do not describe the mechanism(s) that IFN- λ uses to have an antiviral effect but instead state that PEG-IFN- λ has an antiviral effect regardless of sex and describe an interesting finding that IFN- λ still has an anti-viral effect in adults >45 years of age despite a reduced and delayed T cell response. The authors should show the viral load by time data stratified by treatment group and age to support this statement.

Response: We appreciate this suggestion. In fact, we made the claim and had not presented the data. We have added Figure 5 which shows the effect of PEG-IFN- λ versus placebo according to age, above and below 45 y, and discuss this on page 6 line 212-213 and page 8 line 307-310. Patients older than 45y showed the same rate of decline in SARS-CoV-2 RNA as patients <45y.

The new figure has been reviewed by a statistician and they have been added as a co-author to the paper.

The authors also describe the *IFNL4* genotype in their subset of 9 subjects, but there is not a clear connection to their original hypothesis and the authors acknowledge the small sample size.

Response: We analyzed the potential effects of the *IFNL4* genotype on T cell responses in 38 patients and on antibody levels in 27 patients. We did not analyze the genotype in our subset of 9 patients used for scRNAseq. The *IFNL4* genotype is very important in determining treatment outcomes for other IFN sensitive viruses, in particular Hepatitis C virus. *IFNL4* genotype has also been associated with COVID-19 outcomes (Saponi-Cortes, J.M.R. et al. Scientific Reports 2021, Rahimi, P. et al. Virology Journal 2021). Therefore, we felt it was important to attempt the comparison in our study despite the low number of

subjects. We have expanded on why we analyzed the *IFNL4* genotype on page 6, lines 221-223 to provide better background/justification for the reader.

Overall, the findings are of general interest to biopharmaceutical companies, regulators and clinicians that are developing anti-viral therapies for SARS-CoV-2 and other viruses and for basic researchers working to better understand the mechanism(s) of action of IFN- λ . Overall, I found the paper difficult to follow as it is a mix of mechanistic studies and COVID-19 epidemiology comparing age, sex and genetics. The T cell and antibody data is negative and the authors then pivot to showing a collection of interesting observational findings from the study unrelated to the PEG-IFN- λ intervention.

Response: Thank you recognizing and commenting of the broad interest of our work.

Towards making the key finding clearer to the reader a proposed title is “IFN- λ treatment accelerates viral clearance despite not inducing a more robust T cell or antibody response to SARS-CoV-2”. This title more accurately summarizes the key findings of the paper.

Response: As correctly pointed out, our study analyzed both the impact of treatment and epidemiological characteristics on T cell and antibody immunity. While the suggested title captures the effect of IFN- λ on the host immune response, it excludes a key novel finding that T cell responses were delayed at very early time points after infection in older adults but IFN- λ treatment was still effective. Given the more severe impact of Covid19 on older persons, we felt highlighting this observation in the title was important. This observation was further supported in our Phase 3 clinical trial, where IFN- λ reduced hospitalization in high-risk populations. Therefore, we, respectfully, would like to maintain our original title.

Specific Points

B cells, CD8 T cells and pDCs are responsive to PEG-IFN- λ - Figure 1 The group previously reported that functional type III IFN receptors are expressed on human Immune cells, including B and T cell populations (Santer, Minty et al. 2020) and they use single cell RNAseq analysis to show that IFN- λ R mRNA is expressed in CD8 Granzyme B cells, memory B cells and plasmacytoid dendritic cells in 5 subjects that had PEG-IFN- λ and 4 placebo controls. For someone not familiar with single RNA seq the data in Figure 1 is confusing. It would be helpful if the numbers in panel A were transcribed to panels B and C and define what UMAP_1 vs. UMAP_2 is. It appears the authors used a software program to organize the data in panel D and the cell types are listed 0, 1 5, 2, 7, 11... This is very confusing for the reader and it took a long time to connect the numbers in A to the unlabeled pictures in B and C and then to the unordered numbers in D. In the end, the rather simple summary is that B cells, memory B cells, GzB CD8's and pDCs all express IFN- λ receptor RNA as previously reported.

Response: The Seurat package automatically numbers clusters by size irrespective of cell type (Fig 1A). Therefore cluster 0 is the largest cluster and cluster 17 is the smallest. To help clarify the cells within each cluster, we have labelled the clusters with cell types defined in the dot plot, which is now Fig 1B, to the cluster map in Fig 1A,C,D

Figure 1B (previously Fig 1D) is a dot plot intended to define transcript expression in each cluster based on canonical marker genes and are grouped by cell type, ie CD4 T cells, CD8 T cells, NK cells etc. Because there are multiple clusters of CD8 T cells of different sizes, they do not have continuous cluster numbering, ie CD8 T cells are clusters 2, 5, 7, 11, 12 because they contain different numbers of cells. However, CD8 T cells are grouped together on the dot plot to demonstrate their consistent identities, ie all T cells express CD3D (CD3 delta chain) and CD8 T cells express CD8B (CD8 beta chain). If the cell types were aligned by cluster size on the X axis in Fig 1B, the marker genes would be distributed across the dot plot, mixing all the cell types, making it very difficult to see common markers like CD3, CD4 or CD8. We hope that labeling the clusters in Fig 1A,C,D will help resolve some of the confusion which, admittedly, arises when condensing the amount of data in a scRNAseq experiment into 2-dimensional plots.

Uniform Manifold Approximation and Projection-1 (UMAP1) vs. UMAP2 have now been defined in the figure legend. Similarly to tSNE, these axes are arbitrary units that represent the first and second dimensions of the projected low dimensional graph from the high dimensional scRNAseq dataset.

The authors should note whether the maps shown in A, B and C are unique to COVID-19 subjects or whether they are similar to what would be observed in PBMCs from healthy controls or someone with another viral infection (e.g. influenza or HCV).

Respond: The cell types/clusters observed in the COVID-19 patient PBMC is consistent with PBMC from healthy donors. We have added this comment into the results text page 3, lines 91-92. In addition, none of the cell clusters were unique to the acute time point (enrollment, day 0) compared to day 3 and 7 post IFN- λ treatment or placebo controls. Therefore, we are confident we measured transcriptional changes within stable immune cell populations in patient PBMC. Regarding the IFN- λ receptor expression, we, and others, have shown *IFNLR1* mRNA expression in *sorted* immune cells from healthy control PBMCs in similar cell types (Santer PLOS Path 2020, Goel et al PNAS 2020, Coto-Llerena et al. Life Science Alliance 2020).

The last sentence of this section states “Overall, these analysis demonstrated that IFNLR1+ immune cells in the peripheral blood responded to PEG-IFN- λ treatment in COVID-19 patients.” It is not clear, if this is a novel or expected finding. They acknowledge that each of these cell populations were previously demonstrated to respond to IFN- λ in vitro (Yin, Dai et al. 2012, Santer, Minty et al. 2020). The authors should put these current findings in context of what PEG-IFN- λ treatment does to PBMCs in a healthy or virus-infected

individual. Specifically, is there something unique about treating a SARS-CoV-2 infection versus another viral infection with PEG-IFN- λ .

Response: We thank the reviewer for this comment so we can clarify this in our manuscript, which we feel is an important point. Our previous publication showed the IFN- λ receptor is present and functional in specific sorted human peripheral immune cells treated *in vitro* (Santer PLOS Path 2020). Goel et al. PNAS 2020 did add limited scRNA seq data from PBMC of 1 healthy donor, also treated *in vitro*, with IFN- λ . To our knowledge, we are the first to demonstrate sensitivity of specific human immune cell subsets to PEG-IFN- λ treatment *in vivo*. This has been highlighted in the text page 3, line 111-114 and page 7, line 265-267.

This is the first time PEG-IFN- λ has been used to treat an acute viral infection. Previous use of IFN- λ has been restricted to longer term administration in chronic viral infections such as HCV and Hepatitis Delta infection. Therefore, the ability to draw strong conclusions of its effect between infected individuals and healthy donors is limited, especially when PEG-IFN- λ has not been given to uninfected individuals as a control.

Regarding whether there is a unique aspect of treating SARS-CoV-2 infection vs. other viral infections likely relates to the site of infection. IFN- λ may, in fact, serve as a useful treatment for other respiratory viral infections and has been tested in influenza models. However, the success of IFN- λ treatment will likely depend on whether the virus is able to evade or inhibit signaling from IFNLR1. Outside of overexpression models demonstrating potent JAK/STAT signaling inhibition by SARS-CoV-2 proteins, most studies show IFNs inhibit SARS-CoV-2 *in vitro* (eg Vanderheiden et al J Virol 2020, Felgenhauer et al. J Biol Chem 2020, Clementi et al. J Infect Dis 2020, Mantlo et al. Antiviral Res 2020, Metz-Zumaran J Virol 2022)

Recommendations

- **Add labels to B and C**

Response: New figure numbers, Fig 1C,D. We have added labels to all panels

- **Since clusters 2, 6, 10 and 16 (GzB CD8's, B's and pDC's) are key to the authors' hypothesis those should be labeled in B and C.**

Response: All clusters now have labeled and we highlighted clusters of interest using boxes.

- **Organize panel D on the X axis from cell group 0 to 20 rather than by genes on the Y axis.**

Response: Based on the comments above, all of Figure 1 was revamped to have all cell subsets by name and no longer numbered for better clarity. The convention for dot plots is to group clusters by cell type so hopefully labeling the cell types helps to clear up any confusion.

- **Highlight the 4 key IFN- λ R positive cells of interest.**

Response: key cell types have been highlighted using boxes.

- **Label Panels G – J by cell type instead of cluster type.. E.g. instead of Cluster 16 label the graph**

Response: cell names have been added to the graphs.

- **Why show cluster 8 monocytes and not the IFN- λ R positive cluster 2, GzB CD8 cells?**

Response: The use of the monocyte population in Fig 1H was to demonstrate an internal negative control. Monocytes expressed the highest level of the IL-10RB receptor, one component of the IFN- λ receptor but remained unresponsive to IFN- λ treatment, consistent with our *in vitro* studies. This allowed us to state that the maintenance of ISG expression we observed in pDC was specific and not related to endogenous type I IFN that may still be present in patients. We added their use as a control on page 3, lines 106-108 to better clarify this data. CD8 GzmB cells were not displayed because we could not detect an ISG response in the small population of IFNLR1+ enriched cells. This was discussed on page 7, lines 271-277.

Ideally, the authors would also have had SC RNAseq on bronchial epithelial cells (recognize the difficulties in collecting such samples), but they should add text in the discussion of what the ISG response would look like in PEG-IFN- λ treated epithelial cells.

Response: We agree that would have been a nice comparison if samples were feasible to collect, but we and others have demonstrated IFN- λ potently acts on lung epithelial cells to induce ISGs to a greater extent (~10-100 fold) compared to purified immune cells (eg. Santer et al PLoS Path 2020). We have added text to the discussion regarding this response on page 7, lines 277-279.

Antibody levels and antibody responses to SAR-CoV-2 – Figure 2Y axis for the IgM and IgA data should be changed from IgG to IgM and IgA respectively.

Response: Thank you for noting this error. This has been corrected.

The summary sentence should state that “...and PEG-IFN- λ treatment did not inhibit or increase B cell antibody responses measured in plasma.

Response: Thank you for this suggestion. We have changed as suggested page 4, line 143.

PEG-IFN- λ treatment did not affect T cell responses – Figure 3. It would be helpful to have the FluoroSpot data presented on a log scale (assign 0 values to 0.1) and state the percentage of subjects without a discernable T cell response. It appears many of the samples have a value of 0.

Response: As requested, we log transformed and re-analyzed the FluoroSpot data. The Log-transformed graphs can be found at the end of this Response to Reviewers. We tested for normal distribution of the data (Shapiro-Wilkes test) and found that neither the non-transformed nor transformed data were normally distributed, nor did the Log transformation have an impact on our significant observations. Therefore, we retained the original figures using non-transformed data, which is convention for ELISpot/FluoroSpot data. However, if the editor feels the data is better represented using the log-transformed T cell graphs, we will adjust any described statistics and tests based on their decision using the figures below.

The percentages of subjects with a positive T cell response can be found in Supplementary Figure 1.

SARS-CoV-2 specific T cell responses were delayed in older patients – Figure 4 and 5 Figure 5. Add legend describing the different colors in the bar graph (M, spike etc.)

Response: Thank you for this suggestion. We have updated figure legend to describe the colors in Fig 4 and Fig 6 (previously Fig 5)

Line 215: The Acceleration of viral decline with PEG-IFN- λ compared to placebo was not affected by age (OR 0.97, 95% CI 0.93-1.02). This statement is the key foundation to the second half of the paper and the authors should show a graph of viral load versus time stratified by PEG-IFN- λ vs. placebo and <45 vs >45 in the main body of the paper.

Response: As mentioned above, we agree this data should have been added to the manuscript. We have made the graph for viral load decline in patients <45 vs >45y. There was no significant difference in the rate of viral load decline in patients treated with IFN- λ <45 or >45y. Therefore, patients >45y benefited from IFN- λ despite the delayed T cell response. We have included the figure below for easy reference and added Figure 5 to show viral load over time by treatment and by age cut-off.

Clarify which group includes subjects 45 years of age. (\leq or \geq instead of < or >).

Response: No patient was exactly 45y old. This can be seen on the X axis in Figure 6.

Discussion

Line 327. “To summarize, our analysis demonstrates that a single dose with peginterferon accelerates viral clearance without affecting virus specific T cell responses or antibody production in mild to moderate acute SARS CoV-2 infection”. The authors don’t show this data and they should add a graph(s) showing VL overtime stratified by age and treatment group.

Response: Please see response above to address this point.

References

Feld, J. J., C. Kandel, M. J. Biondi, R. A. Kozak, M. A. Zahoor, C. Lemieux, S. M. Borgia, A. K. Boggild, J. Powis, J. McCready, D. H. S. Tan, T. Chan, B. Coburn, D. Kumar, A. Humar, A. Chan, B. O’Neil, S. Noureldin, J. Booth, R. Hong, D. Smookler, W. Aleyadeh, A. Patel, B. Barber, J. Casey, R. Hiebert, H. Mistry, I. Choong, C. Hislop, D. M. Santer, D. Lorne Tyrrell, J. S. Glenn, A. J. Gehring, H. L. A. Janssen and B. E. Hansen (2021). Peginterferon lambda for the treatment of outpatients with COVID-19: a phase 2, placebo-controlled randomised trial. *Lancet Respir Med* 9(5): 498-510.

Santer, D. M., G. E. S. Minty, D. P. Golec, J. Lu, J. May, A. Namdar, J. Shah, S. Elahi, D. Proud, M. Joyce, D. L. Tyrrell and M. Houghton (2020). Differential expression of interferon-lambda receptor 1 splice variants determines the magnitude of the antiviral response induced by interferon-lambda 3 in human immune cells. *PLoS Pathog* 16(4): e1008515.

Yin, Z., J. Dai, J. Deng, F. Sheikh, M. Natalia, T. Shih, A. Lewis-Antes, S. B. Amrute, U. Garrigues, S. Doyle, R. P. Donnelly, S. V. Kotenko and P. Fitzgerald-Bocarsly (2012). Type III IFNs are produced by and stimulate human plasmacytoid dendritic cells. *J Immunol* 189(6): 2735-2745.

Additional References

Saponi-Cortes J.M.R., Rivas M.D., Calle-Alonso F., Sanchez J.F., Costo A., Martin C., Zamorano J. (2021) IFNL4 genetic variant can predispose to COVID-19. *Sci Rep*. <https://doi.org/10.1038/s41598-021-00747-z>

Rahimi, P., Tarharoudi, R., Rahimpour, A., Amroabadi J.M., Ahmadi I., Anvari E., Siadat S.D., Aghasadeghi M., Fateh A. (2021) The association between interferon lambda 3 and 4 gene single-nucleotide polymorphisms and the recovery of COVID-19 patients. *Viol J* . <https://doi.org/10.1186/s12985-021-01692-z>

McInnes L., Healy J. (2018) UMAP: uniform manifold approximation and projection for dimension reduction. *arXiv* <https://arxiv.org/abs/1802.03426v2> [PREPRINT]

Becht E., McInnes L., Healy J., Dutertre C-A, Kwok I.W.H., Ng L.G., Ginhoux F., Newell E.W. (2018) Dimensionality reduction for visualizing single-cell data using UMAP. *Nat Biotechnol* 37: 38–44

Luecken M.D., Theis F.J.. Current best practices in single-cell RNA-seq analysis: a tutorial. *Mol Syst Biol*. (2019) Jun 19;15(6):e8746. Doi: 10.15252/msb.20188746.

Santer D.M., Minty G.E.S., Golec D.P., Lu J., May J., Namdar A., Shah J., Elahi S., Proud D., Joyce M., Tyrrell D.L., Houghton M. (2020) Differential expression of interferon-lambda receptor 1 splice variants determines the magnitude of the antiviral response induced by interferon-lambda 3 in human immune cells. *PloS Pathog*. <https://doi.org/10.1371/journal.ppat.1008515>

Goel R.R., Wang X., O’Neil L.J., Nakabo S., Hasneen K., Gupta S., Wigerblad G., Blanco L.P., Kopp J.B., Morasso M.I., Kotenko S.V., Yu Z-X., Carmona-Rivera C., Kaplan M.J. (2020) Interferon lambda promotes immune dysregulation and tissue inflammation in TLR7-induced lupus. *PNAS*. <https://doi.org/10.1073/pnas.1916897117>

Coto-Llerena M., Lepore M., Spagnuolo J., Di Blasi D., Calabrese D., Suslov A., Bantug G., Duong F.H.T., Terracciano L.M., De Libero G., Heim M.H. (2020) Interferon lambda 4 can directly activate human CD19+ B cells and CD8+ T cells. *Life Sci. Alliance*. <https://doi.org/10.26508/lsa.201900612>

Vanderheiden A., Ralfs P., Chirkova T., Upadhyay A.A., Zimmerman M.G., Bedoya S., Aoued H., Tharp G.M., Pellegrini K.L., Manfredi C., Sorscher E., Mainou B., Lobby J.L., Kohlmeier J.E., Lowen A.C., Shi P-Y., Menachery V.D., Anderson L.J., Grakoui A., Bosinger S.E., Suthar M.S. (2020) Type I and Type III Interferons Restrict SARS-CoV-2 Infection of Human Airway Epithelial Cultures. *J Virol.* <https://doi.org/10.1128/JVI.00985-20>

Felgenhauer U., Schoen A., Gad H.H. Hartmann R., Schaubmar A.R., Failing K., Drosten C., Weber F. (2020) Inhibition of SARS-CoV-2 by type I and type III interferons. *J Biol Chem.* <https://doi.org/10.1074/jbc.AC120.013788>

Clementi N., Farrarese R., Criscuolo E., Diotti R.A., Castelli M., Scagnolari C., Burioni R., Antonelli G., Clementi M., Mancini N. (2020) Interferon- β -1a Inhibition of Severe Acute Respiratory Syndrome-Coronavirus 2 In Vitro When Administered After Virus Infection. *J Infect Dis.* <https://doi.org/10.1093/infdis/jiaa350>

Mantlo E., Bukreyeva N., Maruyama J., Paessler S., Huang C. (2020) Antiviral activities of type I interferons to SARS-CoV-2 infection. *Antiviral Res.* <https://doi.org/10.1016/j.antiviral.2020.104811>

Metz-Zumaran C., Kee C., Doldan P., Guo C., Stanifer M.L., Boulant S. (2022) Increased Sensitivity of SARS-CoV-2 to Type III Interferon in Human Intestinal Epithelial Cells. *J Virol.* <https://doi.org/10.1128/jvi.01705-21>

Reviewer 2

Santer and colleagues report a mechanistic follow-up to their clinical study on use of PEG-IFN- λ in COVID-19 patients, which showed faster viral clearance. The authors conclude that the observed clinical effects were likely the result of ISG induction, as demonstrated by RNAseq showing consistent ISG expression in IFN- λ -responsive immune cell subsets of treated but not placebo-treated patients. At the same time, antibody levels or T cell response were not affected, leading to the conclusion that ISG induction drives virus clearance, which is of importance for treatment of patients with delayed T cell responses such as the elderly. The study is well-conducted and well-written and highly relevant, adding mechanistic data to a recent clinical cohort. Some points should be added to the discussion and some methodological questions need to be clarified. I recommend for publication after minor revision.

Response: Thank you for recognizing our effort to provide mechanistic data to the clinical study. Our goal was to provide a deeper understanding of the *in vivo* effects of IFN- λ in patients, where immunological responses have not been extensively studied.

Specific details:

Line 85: „All 5 patients chosen for scRNAseq analysis who received PEG-IFN- λ demonstrated accelerated virus clearance“ – what about those receiving PEG-IFN- λ who did not demonstrate accelerated virus clearance? ISG induction should work relatively universally – what could lead to reduction of observed clinical effects? High basal IFN- λ and saturation if ISG induction? Please discuss.

Response: Addressing this question could provide valuable insight into the efficacy/responsiveness to interferon treatment. However, only one patient was classified as “non-responder” in the Phase 2 clinical study, which does not provide sufficient power to draw conclusions on the mechanisms of non-response nor was the patient included in scRNAseq analysis. We have re-phrased the above wording in the manuscript so that it no longer sounds like a significant proportion of patients were non-responders and indicated that there was only 1 non-responder in the study page 7, lines 287-290.

Line 104: How was this list of 24 ISGs chosen? There are hundreds of ISGs, how was ensured that this group does lead to biased results? Since the mechanistic conclusions of this manuscript hinges on this result, this needs to be clarified.

Response: We appreciate the careful attention to detail related to the IFN module scores. This list was curated based on the list of known IFN- λ ISGs in our PBMC assay and ISG that were reliably detected and showed a significant change (up or down) from baseline to 3d post- treatment in the 10x Dataset. We added a sentence to the manuscript on page 10, line 415-417 outlining this procedure. We also performed the modular analysis using a list of 155 type I IFN ISG and found a similar result. The gene modules score changes were more modest with the expanded list, as was expected from a larger list focused on type I vs. type III IFN ISG.

Line 107: For Cluster 8 (monocytes), the averages show no elevated ISG response for both groups, but the individual donors analyzed vary considerably and one of the placebo donors has the highest, one of the IFN donors has the lowest response. What is the basis of this, are there individual donor-specific differences in IFN receptor expression in these donors?

Response: Cluster 8, monocytes, were shown because they are known to be non-responsive to IFN- λ *in vitro*. Therefore, they served as an *in vivo* negative control for IFN- λ treatment.

As you have pointed out, there is variability in the module response in monocytes, which is why the data is displayed as change from baseline (indicated in methods), which was within 7d of a positive COVID-19 molecular test and symptom onset. We found that ISG expression was elevated in both placebo and IFN- λ treated patients at baseline because of endogenous type I IFN present during the acute phase, which showed a trend with SARS-CoV-2 viral load at study entry. Because patients were enrolled at different times within the 7 day window, endogenous type I IFN likely declined at different rates between patients as SARS-CoV-2 was cleared, explaining the variability. Therefore, the ISG module continued to decline in monocytes, whereas ISG expression is maintained in IFN- λ responsive pDCs and B cells after treatment.

Line 167: Text refers to Fig 2A, should probably be 3A instead

Response: Thank you for noting this error. It has been corrected.

General: Is there a risk of increased autoimmunity against IFNs when IFN λ is given exogenously? Please discuss.

Response: You are likely aware of the reports by the Casanova group detecting naturally occurring type I IFN auto-antibodies in severe COVID-19 patients. There have also been reports of anti-pegylated IFN- α antibodies in chronic HBV, where patients receive 48w of treatment (Nishio et al. Sci Transl Med 2021). So far, naturally occurring type III IFN auto-antibodies have not been reported in the Casanova et al studies, although a preprint submitted last year suggests the presence of type III IFN auto-antibodies in severe COVID-19 patients (Credle et al. bioRxiv 2021). There have not been reports of anti-IFN- λ antibodies in the context of long-term (48w) PEG-IFN- λ therapy, but the phenomenon has not been extensively studied. It is important to note that our study used a single dose of IFN- λ . Therefore, we anticipate that anti-IFN- λ antibodies are unlikely based on the limited exposure to PEG-IFN- λ . This does not exclude the “risk” of auto-antibodies for PEG-IFN-

λ , which should be considered as a potential mechanism for non-response to therapy, particularly in any future longer-term treatments.

What about levels of endogenous IFN- λ pre administration of the exogenous IFN- λ ? Given that ISG levels did not further increase in the IFN group but just stayed high consistently, might this contain clues on when IFNs would be ideally administered? Possibly better at a later timepoint, when infection-induced IFNs drop? Please discuss.

Response: We did perform Luminex on plasma samples at study entry for both type I and type III IFNs but failed to detect significant levels in the periphery by the time patients were enrolled. For reference, patients were enrolled within 7d of symptom onset, which would likely be 9d post-infection, and quite late to detect endogenous IFNs in the periphery in mild COVID19 patients.

Regarding the second point of maintaining ISG expression rather than increasing expression, please refer to the point above regarding ISG expression in monocytes. ISG were induced by SARS-CoV-2 infection, indicating that IFN is/was recently present in these patients. This was discussed on page 7, lines 279-283. A key feature of type III IFNs is that their ISG induction is more long-lived because they are not subject to a strong negative feedback loop or desensitization induced by IFN- α for example.

There may be opportunities to optimize the timing of administration. This has been tested in animal models and we have used a multiple dosing strategy in hospitalized COVID-19 patients (work in progress). However, the same single dosing strategy, within 7d of symptom onset, was highly successful, reducing hospitalization by 60% in high-risk populations, in the Phase 3 clinical trial, suggesting that early administration is key to achieve the beneficial effect.

Additional References

Nishio A., Bolte F.J., Takeda K., Park N., Yu Z-X, Park H., Valdez K., Ghany M.G., Rehermann B. (2021). Clearance of pegylated interferon by Kupffer cells limits NK cell activation and therapy response of patients with HBV infection. *Sci Transl Med*. <https://doi.org/10.1126/scitranslmed.aba6322>

Credle J.J., Gunn J., Sangkhapreecha P., Monaco D.R., Zheng X.A., Tsai H-J, Wilbon A., Morgenlander W.R., Dong Y., Jayaraman S., Tosi L., Parekkadan B., Baer A.N., Roederer M., Bloch E.M., Tobian A.A.R., Zyskind I., Silverberg J.I., Rosenberg A.Z., Cox A.L., Lloyd T., Mammen A.L., Larman B. (2021). Neutralizing IFNL3 Autoantibodies in Severe COVID-19 Identified Using Molecular Indexing of Proteins by Self-Assembly. *bioRxiv*. <https://doi.org/10.1101/2021.03.02.432977> [PREPRINT]

Log Transformed Fluorospot Graphs

Figure 3. Comparison of T cell responses between placebo and PEG-IFN- λ treated COVID-19 patients at day 0, day 7, and day 90+ post enrollment. A) IFN-g B) IL-2 C) Polyfunctional (IFN-g + IL-2) T cell responses (as SFUs per 10^6 PBMCs) against structural SARS-CoV-2 protein peptide pools were quantified ex vivo using FluoroSpot assays. Each dot represents a different patient. Only significant differences between time points are shown where * = $p < 0.05$, ** = $p < 0.01$ (Wilcoxon signed-rank test). Bar lines represent median and 95% CI.

Figure 4. Differences in T cell responses between patients below and above the median age at day 0, day 7, and day 90+ post-enrollment. A) IFN-g B) IL-2 C) Polyfunctional (IFN-g + IL-2) T cell responses (as SFUs per 10^6 PBMCs) against structural SARS-CoV-2 protein peptide pools were compared between patients above and below the median age of the whole cohort (45 years old). Each dot represents a different patient. Only significant differences between groups are shown where * = $p < 0.01$, *** = $p < 0.001$ (Mann-Whitney U-tests). Bar lines represent median and 95% CI.

A. IFN- γ

B. IL-2

C. IFN- γ + IL-2

Supplementary Figure 7. Differences in T cell responses between sex at day 0, day 7, and day 90+. A) IFN-g B) IL-2 C) Polyfunctional (IFN-g + IL-2) T cell responses (as SFUs per 10^6 PBMCs) against structural SARS-CoV-2 protein peptide pools were compared between male and females. Significant differences were not observed using Mann-Whitney U-tests between sexes ($p < 0.05$). Bar lines represent median and 95% CI.

Supplementary Figure 8. Comparison of T cell responses between IFNL4 genotype at day 0, day 7, and day 90+ post-enrollment. A) IFN-g B) IL-2 C) Polyfunctional (IFN-g + IL-2) T cell responses (as SFUs per 10^6 PBMCs) against structural SARS-CoV-2 protein peptide pools were compared between genotypes. “ $_G$ ” indicates non-TT rs368234815 polymorphisms at the IFNL4 locus. Significant differences were not observed using Mann-Whitney U-tests between genotypes ($p > 0.05$). Bar lines represent median and 95% CI.

REVIEWERS' COMMENTS

Reviewer #1 (Remarks to the Author):

Summary:

The authors have satisfactorily addressed the questions and concerns and I find the paper acceptable for publication.

Reviewer #2 (Remarks to the Author):

The authors have answered my open questions and expanded the discussion where necessary. The manuscript is consequently improved and I have no further criticisms and recommend for publication.